# Practical Memorization Tests for Detecting Copyrighted Data in Large Language Models

## Abstract

Large language models (LLMs) have been shown to reproduce copyrighted text, e.g., passages from news articles, sparking high-profile lawsuits against their providers. Yet copyright claims often rely on anecdotal evidence. To strengthen such cases, there is a clear need for automated, interpretable, and reliable methods to detect repetitions that are also suitable in legal proceedings. Current methods often equate an LLM's ability to complete prefixes from source passages with memorization, but this metric fails when strong completions arise from generalization. We identify four key criteria for practical real-world deployment: methods must operate with only black-box access to the LLM, while remaining reliable (high precision), efficient, and interpretable. We first investigate prior work and find simple non-memorized counterexamples that trigger false positives even in advanced methods designed to account for generalization, undermining their reliability in legal contexts. Going further, we introduce **DualTest**, which requires only a single API call per passage under test and produces human-interpretable results. Our key insight is to leverage a small proxy model to disentangle memorization from generalization in counterexamples, improving recall by up to 3% compared to the strongest baseline.

## 1 Introduction

Training Large Language Models (LLMs) requires massive datasets, and, as high-quality public data on the Internet has largely been exhausted, LLM providers are increasingly harnessing copyrighted data in their training, sometimes liillicit means (ArentFox Schiff Law Firm, 2025). Such cases raise legal concerns regarding whether copyrighted data can be utilized in LLM training, without a permissive licensing agreement (Robertson, 2024). Content creators have already challenged this practice in court through several high-profile lawsuits naming LLM providers as defendants (Getty Images, 2023). However, these cases have not always been decided in the creators' favor. For instance, in *Kadrey v. Meta Platforms, Inc.*, the judge declined to rule on the legality of training on copyrighted materials and dismissed claims that Meta's models would cause financial harm to authors. Similarly, in *Bardz v. Anthropic*, the court held that Anthropic's use of copyrighted books for training was permissible, reasoning that the models' outputs were transformative and therefore protected under fair use (ArentFox Schiff Law Firm, 2025).

These court cases highlight that while LLMs may be trained on legally obtained copyrighted content, their outputs must be sufficiently transformative to avoid resembling the original work (The National Law Review, 2025). Accordingly, we need methods that can determine both **(A)** whether a suspect LLM was trained on a given text sample, and **(B)** whether its outputs are non-transformative relative to that sample. In court settings, each side actively seeks to discredit the methods relied upon by the opposition to strengthen its own case. Consequently, it is paramount that any method used to support LLM copyright violation claims be robust against deliberate attempts to undermine its validity.

Past membership inference attacks (MIAs) have shown promise in detecting whether specific samples were memorized during training, for instance, by measuring the likelihood of the target model reproducing a sample (with memorized samples exhibiting lower perplexity). Unfortunately, such methods cannot reliably distinguish between memorization and generalization. For example, when

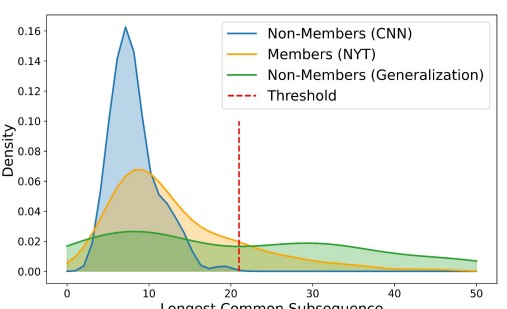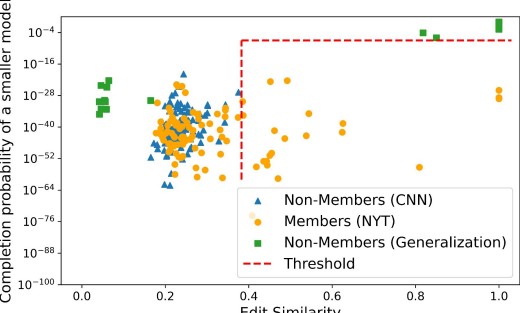

Figure 1: (Left) Prior MIA (Karamolegkou et al., 2023) relies on the longest common subsequence (LCS) between source text and LLM output, but can produce false positives when LLMs reproduce non-member texts via generalization. (Right) DUALTEST leverages a smaller reference LLM to disentangle memorization from generalization, enabling reliable detection in real-world scenarios.

a sample is highly repetitive, an LLM may correctly reproduce the remainder from a prefix, even without having memorized it. In addition to producing false positives (erroneously marked as memorized), such *counterexamples* can be trivially constructed and weaponized by defendants to challenge and discredit MIAs as valid evidence. Note that these counterexamples must exhibit clear, legible repetitive patterns and are distinct from prompts deliberately optimized to elicit specific outputs from an LLM (Schwarzschild et al., 2024), which typically appear random and are therefore less likely to be accepted as credible evidence.

To handle such counterexamples, prior work has proposed augmented MIAs that incorporate sample compressibility, typically measured using the `Zlib` algorithm, to avoid falsely labeling highly repetitive samples as members (Carlini et al., 2021; Kaneko et al., 2024). However, we show that it remains trivial to construct counterexamples even against Zlib-enhanced tests, such as repetitive text with longer cycle lengths. As a result, no existing test currently satisfies both **(A)** and **(B)** while remaining robust to a broader class of counterexamples.

To address this gap, we outline four essential properties for a practical, court-viable test of LLM copyright violations. Such a test must: (i) require only black-box access to the suspect LLM, to remain viable even against non-compliant defendants; (ii) be reliable (high precision with negligible false positives), including against advanced counterexamples; (iii) be efficient enough to scale across large sets of target samples; and (iv) yield human-interpretable results that can serve as credible supporting evidence in legal proceedings.

We propose **DUALTEST** to meet these criteria and overcome the challenges faced by prior approaches. DUALTEST operates in fully black-box settings, requiring only a single API call to the suspect LLM per sample. It remains reliable against both simple and advanced counterexamples, while producing interpretable results, e.g., returning a probability or a similarity score when the test is positive, rather than the opaque quantities produced by prior approaches.

We validate our method through extensive experiments on three settings: (i) proprietary models such as GPT-4, using samples from a real court case (Ravichander et al., 2025); (ii) open-weight, open-dataset LLMs (Biderman et al., 2023), where ground-truth training members are available and in-distribution non-members can be collected; and (iii) popular open-weight but closed-dataset LLMs (Touvron et al., 2023). DUALTEST outperforms all prior baselines in both controlled experiments on Pythia models and evaluations on state-of-the-art models such as GPT-4, achieving consistently higher recall. Furthermore, when applied to LLaMA-2-70B on one million news articles, DUALTEST detects hundreds of memorized samples, demonstrating its scalability and applicability in large-scale settings. Our findings lay the foundation for LLM training data auditing tools that are practical to use in scenarios such as legal proceedings. In summary, this paper makes the following contributions:

- We define four key criteria for practical membership inference attacks, aligned with current legal precedents (training on copyrighted data is permissible, but reproducing it is not): black-box feasibility, high precision, efficiency, and interpretability.

- We propose DUALTEST, a dual-model black-box method that meets all four criteria. DUALTEST uses a small reference model to separate memorization from generalization and introduces two single-query, interpretable detection strategies: run-length and edit-similarity.
- We evaluate DUALTEST on over *1M* news articles (NYT, Guardian, BBC) with open-source LLMs, and through case studies on proprietary models like GPT-4, showing that it achieves high recall at 100% precision in both controlled and real-world settings.

## 2 RELATED WORK

We briefly review prior work on Membership Inference Attacks, Content Memorization, and LLM Copyright Infringement. A more detailed discussion of related work is provided in the appendix.

**Memorization and Membership Inference in Language Models.** Discoverable memorization investigates the memorization of predetermined (target) samples by language models (Carlini et al., 2022b). Schwarzschild et al. (2024) introduces a metric for quantifying memorization based on the minimal prefix length required to generate verbatim outputs. Carlini et al. (2022c) demonstrate that memorization is influenced by data distribution, noting that outlier samples are memorized more easily. Tirumala et al. (2022) observe that memorization can emerge even before models reach the stage of overfitting. Jagielski et al. (2022); Kiyomaru et al. (2024) highlight that earlier training samples are more prone to forgetting. Zhang et al. (2023) examine the impact of memorization on model predictions at test time, while Golchin et al. (2024) explore the relationship between memorization and in-context learning.

Prior work has revealed that preventing verbatim memorization during generation alone is insufficient for ensuring privacy, as approximate memorization can still occur, leading to outputs that are similar to training data (Ippolito et al., 2022).

**Copyright.** Xu et al. (2024) show that LLMs are not effective at respecting the copyright status of source texts. Freeman et al. (2024) present a case study examining large models in the context of the New York Times v. OpenAI lawsuit. Mezzi et al. (2025) explore the gap between existing legal frameworks and the realities of generative AI. Several works discuss the limitations of Fair Use in addressing copyright issues related to generative AI (Rahman & Santacana, 2023; Henderson et al., 2023; Rodriguez Maffioli, 2023). Mueller et al. (2024) evaluate and compare state-of-the-art models using copyright violation metrics grounded in European copyright law. Casper et al. (2024) highlight the advantages of white-box audits over black-box evaluations for AI accountability.

A number of works investigate prefix probing-based methods for detecting copyrighted content in LLM outputs (Karamolegkou et al., 2023; Zhao et al., 2024; Ravichander et al., 2025). Tan et al. (2024) propose LLM-based methods to evaluate potential copyright infringements. Li et al. (2024) introduce an optimization-based framework for identifying copyrighted material in training data, while Duarte et al. (2024) develop a document-level detection approach based on a model's ability to differentiate original from paraphrased text. Prefix probing and optimization-based techniques focus on extracting memorized sequences, but they often require multiple queries and do not distinguish memorization from generalization as clearly as our proxy-based approach.

**Challenges and Evaluation Issues regarding MIAs.** Recent work has highlighted problems in how MIAs are evaluated. Meeus et al. (2024) review existing MIA benchmarks and find that many suffer from distribution shifts between member and non-member datasets, which are often created post-hoc without proper randomization. This can lead to misleading conclusions about a model's memorization. Das et al. (2024) similarly show that due to these distribution shifts, simple "blind" baselines that do not use model outputs can outperform state-of-the-art MIAs. They argue that current evaluation methods may not truly capture membership leakage.

While prior work has made significant progress in understanding memorization and membership inference in LLMs, these methods often require privileged model access, rely on large numbers of queries, or lack interpretability—rendering them impractical in legal or regulatory settings. In contrast, **our work is the first to design MIAs that meet all four practical requirements**: black-box access, high precision, single-query efficiency, and human interpretability. We also focus explicitly on copyright auditing, a setting with distinct legal and evidentiary demands.

## 3    METHOD

We aim to test whether a black-box large language model (the *target* model) has memorized a given copyrighted text sample and can reproduce it closely, even if not perfectly.

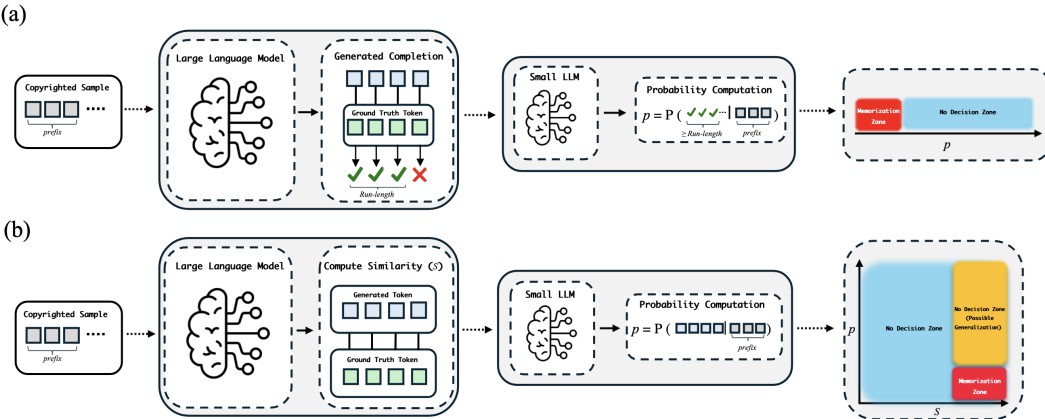

Figure 2: **(a) Illustration of DUALTEST (*Run-Length-Based*).** First, we prompt the target LLM with a 64-token prefix from the copyrighted source document. Next, we count the number of tokens (called the run-length) in the generated completion that *exactly* match the remaining source tokens before divergence. We then compute the probability of generating the same source tokens with this run-length (or longer) under a small reference model. Low likelihood scores from the reference model, combined with high run-length, are interpreted as strong signals of memorization. **(b) Illustration of DUALTEST (*Edit-Similarity-Based*).** First, we prompt the target LLM with a 64-token prefix from the copyrighted source document. Next, we compute the edit similarity between the completion and the ground truth (maximum 64 tokens). We then compute the probability of generating the same completion by a small reference model. High similarity by the target LLM combined with low probability by the reference model are interpreted as strong signals of memorization.

### 3.1    REQUIREMENTS FOR A PRACTICAL MEMORIZATION TEST

**(1) Non-Privileged Access.** Considering uncooperative LLM providers, the memorization tests should rely only on model outputs (generated tokens) observable through the LLM's public API. Existing implementations and LLM protections (e.g., against model stealing (Carlini et al., 2023)) preclude access to privileged information, such as detailed log-probabilities relied on by some prior work (Xie et al., 2024).

**(2) Reliability.** A viable MIA must allow tuning to obtain exceptionally high precision even in adversarial settings. Previous work has shown that membership tests that fail to distinguish between true memorization and generalization become fragile (Tramèr et al., 2022). In legal settings, LLM providers are motivated to exploit this by presenting convincing non-member counterexamples that falsely trigger the test, undermining its evidentiary value. Later, we will show how to generate such examples, highlighting the adversarial nature of the problem that prior work did not address.

**(3) Efficiency.** Many content creators have massive archives (e.g., the NYT archives have over 13M articles), and they cannot know in advance where to start testing a suspect LLM. In this setting, methods that require extensive API interactions per sample are prohibitively costly and can look like forced extraction rather than detection.

**(4) Interpretability.** For use in legal contexts, the detection criteria must be clear, transparent, and accessible to non-expert audiences (e.g., a judge). Complex metrics lacking straightforward interpretability (such as perplexity or BLEU scores) undermine the practical viability of a test.

## 3.2 DESIGNING DUALTEST

We present DUALTEST, a membership test designed to satisfy the four real-world desiderata outlined above. At a high level, DUALTEST prompts the suspect (target) LLM with a fixed-length prefix from a known copyrighted source text and evaluates the sequence of completion tokens the LLM generates. Our goal is to tell apart outputs caused by memorization from those caused by normal language modeling. To do this, we use a smaller, open-source *reference model* as a proxy for generalization. If the target's output is very close to the source continuation and the reference model assigns low probability to producing something that close, we treat that as evidence of memorization.

We implement two complementary variants: a *run-length-based* variant (exact matches) and an *edit-similarity-based* variant (approximate matches). Figure 2(a) sketches the run-length-based (RLB) variant; Figure 2(b) sketches the edit-similarity-based (ESB) variant.

## 3.3 RUN-LENGTH-BASED DUALTEST

From each source document, we take a 64-token prefix and ask the target model for up to 64 completion tokens. The **run length** is the number of consecutive completion tokens that exactly match the source continuation, stopping at the first mismatch.

To judge whether a long run reflects memorization or just generalization, we estimate how likely it is that the *reference* model would produce a run at least that long on the same prefix. A long run that is very unlikely under the reference model suggests memorization by the target.

## 3.4 EDIT SIMILARITY-BASED DUALTEST

Prior work suggests that verbatim memorization is more easily prevented than approximate memorization (Ippolito et al., 2022). To detect *approximate* memorization, we use an edit-based variant. We again prompt with a 64-token prefix and take up to 64 completion tokens from the target.

Instead of exact matches, we compute **edit similarity** between the target completion and the source continuation (Ippolito et al., 2022).

As in the run-length variant, we use the reference model to estimate the probability of generating the same completion as the target LLM given the source prefix. A high edit similarity—combined with low probability under the reference model—serves as strong evidence that the target LLM has memorized the ground-truth completion, rather than merely generalizing from linguistic patterns.

## 3.5 WHY DUALTEST MEETS THE FOUR REQUIREMENTS

**Non-Privileged Access.** DUALTEST uses only generated tokens from standard API calls.

**Reliability.** We set thresholds on (i) edit similarity and (ii) the reference-model likelihood. This way we can raise precision and reduce false positives.

**Efficiency.** Each sample needs one short prompt (64-token prefix) and one short completion (up to 64 tokens) from the target, plus runs on a small open-source reference model. This makes it feasible to test very large archives.

**Interpretability.** The decision rule is easy to state: "Did the model continue the source for enough tokens, or very closely, in a way the reference model would almost never do?"

# 4 EVALUATION

We evaluate our approaches in both controlled ("normal") and adversarial (court-like) settings.

## 4.1 EVALUATION SETUP

**Models.** Our main experiments use Pythia-12B as the *target model*. We use smaller models from the Pythia family—410M, 1B, 1.4B, and 2.8B—as the *reference models*. We additionally report results using GPT-4 (Achiam et al., 2023) (with LLaMA-3.1-8B, LLaMA-3.1-70B, and LLaMA-3.1-405B

as references) in the appendix and LLaMA-2-70B (Touvron et al., 2023) (with LLaMA-2-7B as reference).

**Datasets (normal setting).** For our main evaluation, we use the MIMIR benchmark (Duan et al., 2024), which avoids issues of distribution shift and provides clean member/non-member splits. For LLaMA-2-70B, we include results on other datasets with samples from the New York Times, BBC, and The Guardian. These experiments also include large-scale runs on over 1 million articles.

**Generalization Sets (adversarial setting).** To simulate adversarial conditions typical in court settings, we construct *generalization sets*: texts that can be easily generated through pattern completion but are not in the training set (i.e., *counterexamples*). **Generalization Set A** contains 100 very simple and highly repetitive samples (e.g., "Part I — Part II — Part III — Part IV — ..." inspired by (Liu et al., 2025)). **Generalization Set B** contains 10 harder cases with longer cycles and less obvious repetition. To ensure these texts are not part of the target LLM's training data, we manually verify their absence from the web by searching for exact matches online.

**Baselines.** We consider query-efficient, black-box MIAs, categorized into three groups:

**(1) Binary Metric-Based Methods.** These methods produce a binary decision on whether a sample was memorized. Kiyomaru et al. (2024) introduce two such methods: exact match (EM) and a threshold on BLEU score. Similarly, Zhao et al. (2024) apply a fixed threshold on ROUGE-L. To enable precision-controlled comparisons (evaluating recall at fixed precision levels), we adapt these methods to produce continuous scores: **(a)** For EM, we compute the run length: the number of consecutive tokens that exactly match the ground truth. **(b)** For BLEU and ROUGE-L, we use raw similarity scores (before binarization). We mark the updated methods by (+) in the tables.

**(2) Continuous Metric-Based Methods.** Karamolegkou et al. (2023) utilize two similarity-based methods: longest common subsequence (LCS) and normalized Levenshtein distance, which yield continuous scores that enable flexible threshold tuning to support precision-controlled evaluation.

**(3) Less Efficient Methods.** Some recent black-box methods, such as those by Ravichander et al. (2025), He et al. (2025), and Kaneko et al. (2024), run multiple queries per sample. We include a single-query variant of Kaneko et al. (2024) in our evaluation. Their method uses ROUGE-1 (with or without Zlib compression). Analyses of the computational costs of the original version and of the other baselines are included in the supplementary material.

**Evaluation Metrics.** We evaluate using three key metrics that are most relevant in practice.

**(1) Precision (controlled/normal setting):** Fraction of samples flagged as memorized that are true members.

**(2) Recall:** Fraction of true members that are correctly flagged.

**(3) False Positive Rate (FPR, real-world/adversarial setting):** Fraction of generalization samples incorrectly flagged as memorized.

We highlight that precision is computed in the normal setting (without any intervention with the generalization sets), while FPR is computed in the adversarial setting (on the generalization sets). This reflects the legal scenario: if a copyright holder claims high precision for the test's results, an LLM provider could challenge this in court by presenting adversarial counterexamples—texts that the test flags as memorized (potentially even by any state-of-the-art model, not just the target) but that are in fact not memorized.

## 4.2 RESULTS

We evaluate prior binary and continuous metric-based MIAs, as well as our approaches, across seven datasets from MIMIR under both normal and adversarial evaluation settings.

### 4.2.1 BINARY METRIC–BASED METHODS

Table 1 shows the performance of binary metrics such as Kiyomaru et al. (2024) (EM), Kiyomaru et al. (2024) (BLEU), and the method from Zhao et al. (2024). These methods achieve perfect or near-perfect precision across datasets, but recall is frequently 0%. An exception is GitHub, where

recall reaches 33–47%. This indicates that while such metrics are highly conservative, they often fail to detect memorized samples.

Their weakness becomes more apparent when evaluated under adversarial conditions. Table 2 shows false positive rates on generalization set A, where all binary metrics suffer from extremely high FPR (86–88%). Thus, despite high precision in normal settings, these methods break down when tested against adversarially challenging examples.

| Method | Pile-CC | | ArXiv | | DM Math | | GitHub | | HackerNews | | PMC | | Wikipedia (en) | |
|---|---|---|---|---|---|---|---|---|---|---|---|---|---|---|
| | Prec | Rec | Prec | Rec | Prec | Rec | Prec | Rec | Prec | Rec | Prec | Rec | Prec | Rec |
| Kiyomaru et al. (EM) | 100% | 0% | 100% | 0% | 100% | 0% | 98.9% | 33.6% | 100% | 0% | 100% | 0% | 100% | 0% |
| Kiyomaru et al. (BLEU) | 100% | 0.4% | 100% | 0% | 100% | 0% | 98.4% | 45.5% | 100% | 0% | 100% | 0% | 100% | 0.3% |
| Zhao et al. | 100% | 0.5% | 100% | 0% | 100% | 0% | 99.2% | 47.4% | 100% | 0% | 100% | 0% | 100% | 0.4% |

Table 1: Precision and recall of binary metric–based methods across datasets in the normal setting. While precision is consistently ∼100%, recall is almost always 0%, indicating these methods are overly conservative.

| Method | Kiyomaru et al. (2024) (EM) | Kiyomaru et al. (2024) (BLEU) | Zhao et al. (2024) |
|---|---|---|---|
| **FPR** | 86.0% | 88.0% | 88.0% |

Table 2: False positive rates of binary metrics on generalization set A. Despite high precision in normal settings, these methods fail under adversarial setting evaluation with FPRs above 86%.

### 4.2.2 CONTINUOUS METHODS

We next evaluate continuous similarity-based methods, as well as our continuous adaptations of binary metrics (denoted by (+)). Results in Table 3 tune all methods for 100% precision and report recall alongside FPR. Most methods achieve non-zero recall but incur substantial FPR. The only exception is Kaneko et al. (2024) with Zlib compression, which achieves up to 48% recall on GitHub while maintaining 0% FPR.

Table 4 further enforces both 100% precision in the normal setting and 0% FPR in the adversarial setting (with the generalization set A). Here, all methods collapse to 0% recall except Kaneko et al. (2024) with Zlib, which achieves non-zero recall on 5 out of 7 datasets. However, this approach fails under the more difficult adversarial setting when the generalization set B is included (Table 5), where recall is reduced to 0% for 6 out of 7 datasets.

| Method | Pile-CC | | ArXiv | | DM Math | | GitHub | | HackerNews | | PMC | | Wikipedia (en) | |
|---|---|---|---|---|---|---|---|---|---|---|---|---|---|---|
| | Rec | FPR | Rec | FPR | Rec | FPR | Rec | FPR | Rec | FPR | Rec | FPR | Rec | FPR |
| Kiyomaru et al. (EM)(+) | 0.4% | 88.0% | 1.6% | 91.0% | 3.4% | 91.0% | 0.0% | 0.0% | 0.8% | 88.0% | 0.8% | 88.0% | 1.9% | 91.0% |
| Kiyomaru et al. (BLEU)(+) | 0.6% | 88.0% | 4.0% | 91.0% | 4.5% | 88.0% | 0.0% | 0.0% | 0.2% | 88.0% | 0.8% | 88.0% | 2.5% | 88.0% |
| Zhao et al. (+) | 0.7% | 88.0% | 2.0% | 88.0% | 1.1% | 88.0% | 0.0% | 0.0% | 1.1% | 88.0% | 1.0% | 88.0% | 1.5% | 88.0% |
| Karamolegkou et al. (LCS) | 0.4% | 18.0% | 0.2% | 72.0% | 2.2% | 41.0% | 23.1% | 14.0% | 0.0% | 74.0% | 1.0% | 74.0% | 1.2% | 51.0% |
| Karamolegkou et al. (Leven.) | 0.5% | 95.0% | 2.8% | 98.0% | 11.2% | 97.0% | 0.0% | 0.0% | 0.8% | 98.0% | 0.0% | 95.0% | 0.6% | 95.0% |
| Kaneko et al. | 0.8% | 88.0% | 1.8% | 88.0% | 0.0% | 88.0% | 0.0% | 0.0% | 1.1% | 88.0% | 1.0% | 88.0% | 1.8% | 88.0% |
| Kaneko et al. (w/ Zlib) | 0.4% | 0.0% | 1.2% | 1.0% | 6.7% | 11.0% | 48.5% | 0.0% | 0.3% | 8.0% | 1.2% | 1.0% | 2.9% | 0.0% |

Table 3: Performance of continuous methods (including adapted binary metrics). Methods are tuned for 100% precision. Most approaches trade small amounts of recall for high FPR; only the Zlib-enhanced method from Kaneko et al. (2024) achieves meaningful recall while maintaining 0% FPR.

### 4.2.3 OUR METHODS

We compare against this strongest baseline in the adversarial setting with the generalization sets A and B. As shown in Table 5, our methods achieve non-zero recall across six out of seven datasets, with improvements of up to 3% absolute recall on GitHub. This demonstrates that under strict constraints (100% precision in the normal setting, 0% FPR in the difficult adversarial setting), our approach retains meaningful detection capability while existing baselines collapse.

To better understand the role of the small reference model, we augment the Zlib-based baseline with Pythia models of different sizes (in our framework, i.e., we use their metric as similarity) The results

| Method | Pile-CC | ArXiv | DM Math | GitHub | HackerNews | PMC | Wikipedia (en) |
|---|---|---|---|---|---|---|---|
| | Rec | Rec | Rec | Rec | Rec | Rec | Rec |
| Kiyomaru et al. (EM)(+) | 0.0% | 0.0% | 0.0% | 0.0% | 0.0% | 0.0% | 0.0% |
| Kiyomaru et al. (BLEU)(+) | 0.0% | 0.0% | 0.0% | 0.0% | 0.0% | 0.0% | 0.0% |
| Zhao et al. (+) | 0.0% | 0.0% | 0.0% | 0.0% | 0.0% | 0.0% | 0.0% |
| Karamolegkou et al. (LCS) | 0.0% | 0.0% | 0.0% | 0.0% | 0.0% | 0.0% | 0.0% |
| Karamolegkou et al. (Leven.) | 0.0% | 0.0% | 0.0% | 0.0% | 0.0% | 0.0% | 0.0% |
| Kaneko et al. | 0.0% | 0.0% | 0.0% | 0.0% | 0.0% | 0.0% | 0.0% |
| Kaneko et al. (w/ Zlib) | 0.4% | 0.6% | 0.0% | 48.5% | 0.0% | 1.2% | 2.9% |

Table 4: Recall of continuous methods tuned for both 100% precision and 0% FPR on generalization set A. All methods collapse to 0% recall except the baseline that uses Zlib, which recovers up to 48.5% recall on GitHub.

are shown in Table 6. Recall improves by 0.3–8.4% compared to the case without a small reference model, confirming that model-based reference signals complement compression-based heuristics.

Finally, Table 7 shows our upgraded method that combines edit similarity with the length of Zlib-compressed text. This modification improves robustness against repetitive non-members that would yield to a too conservative similarity threshold (we found such examples in the Github dataset) and yields non-zero recall on all datasets. The gains are especially significant on GitHub, where recall increases by 8.4% with Pythia-2.8B as the reference model.

| Method | Pile-CC | ArXiv | DM Math | GitHub | HackerNews | PMC | Wikipedia (en) |
|---|---|---|---|---|---|---|---|
| | Rec | Rec | Rec | Rec | Rec | Rec | Rec |
| Kaneko et al. (w/ Zlib) | 0.0% | 0.0% | 0.0% | 0.4% | 0.0% | 0.0% | 0.0% |
| DUALTEST (RLB) (410M) | 0.0% | 0.0% | 0.0% | 0.7% | 0.0% | 0.0% | 0.1% |
| DUALTEST (RLB) (1B) | 0.0% | 0.0% | 0.0% | 0.4% | 0.0% | 0.0% | 0.0% |
| DUALTEST (RLB) (1.4B) | 0.3% | 0.0% | 0.0% | 1.1% | 0.3% | 0.0% | 0.0% |
| DUALTEST (RLB) (2.8B) | 0.3% | 0.4% | 0.0% | 3.4% | 0.3% | 0.0% | 0.0% |
| DUALTEST (ESB) (410M) | 0.2% | 2.0% | 2.2% | 0.0% | 0.8% | 0.0% | 0.4% |
| DUALTEST (ESB) (1B) | 0.0% | 2.8% | 2.2% | 0.0% | 0.8% | 0.0% | 0.5% |
| DUALTEST (ESB) (1.4B) | 0.0% | 2.6% | 2.2% | 0.0% | 0.8% | 0.0% | 0.3% |
| DUALTEST (ESB) (2.8B) | 0.3% | 3.6% | 2.2% | 0.0% | 0.9% | 0.0% | 0.8% |

Table 5: Recall under the stricter adversarial setting (with generalization sets A and B). The Zlib baseline collapses, while our methods achieve non-zero recall on most datasets (6 out of 7), demonstrating superior robustness.

| Method | Pile-CC | ArXiv | DM Math | GitHub | HackerNews | PMC | Wikipedia (en) |
|---|---|---|---|---|---|---|---|
| | Rec | Rec | Rec | Rec | Rec | Rec | Rec |
| Kaneko et al. (w/ Zlib) | 0.0% | 0.0% | 0.0% | 0.4% | 0.0% | 0.0% | 0.0% |
| +Pythia-410M | 0.4% | 1.0% | 1.1% | 7.5% | 0.3% | 0.4% | 2.8% |
| +Pythia-1B | 0.1% | 1.0% | 4.5% | 3.4% | 0.3% | 0.4% | 1.8% |
| +Pythia-1.4B | 0.4% | 1.2% | 6.7% | 7.1% | 0.3% | 1.2% | 2.8% |
| +Pythia-2.8B | 0.4% | 1.2% | 6.7% | 9.0% | 0.3% | 1.2% | 2.8% |

Table 6: Effect of augmenting the Zlib baseline with small Pythia reference models (we use our similarity-based framework). Recall improves across datasets, showing that model-based reference complements compression heuristics.

### 4.3 Additional Results on LLaMA-2-70B and GPT-4

We evaluate LLaMA-2-70B as the target model, using LLaMA-2-7B as a smaller reference model, across more than 1 million samples. In Table 8 we report recall under a setup where samples from the BBC (from 2025) are treated as non-members for the normal setting, while the generalizable samples are treated as non-members for the adversarial setting. All results are tuned for 100% precision in normal settings and 0% FPR in adversarial settings. It is important to note that while the recall for sources like The New York Times (NYT) may appear low, it still corresponds to the detection of hundreds of memorized articles.

| Small Model | Pile-CC | ArXiv | DM Math | GitHub | HackerNews | PMC | Wikipedia (en) |
|---|---|---|---|---|---|---|---|
| | Rec | Rec | Rec | Rec | Rec | Rec | Rec |
| Pythia-410M | 0.4% | 1.0% | 0.0% | 6.1% | 0.6% | 0.0% | 1.6% |
| Pythia-1B | 0.0% | 1.4% | 2.2% | 2.7% | 0.6% | 0.0% | 1.3% |
| Pythia-1.4B | 0.4% | 1.6% | 3.4% | 6.1% | 0.6% | 0.8% | 2.4% |
| Pythia-2.8B | 0.4% | 1.6% | 3.4% | 8.4% | 0.6% | 0.8% | 2.5% |

Table 7: An enhanced version of our method combining edit similarity with Zlib-compressed length of the text. This modification yields non-zero recall across all datasets, with strongest gains on GitHub.

We include additional results on GPT-4 in the appendix.

| BBC 2017-2023 | Guardian 2016-2018 | NYT 1988-1995 | NYT 2000-2007 |
|---|---|---|---|
| 0.83% | 0.43% | 0.01% | 0.10% |

Table 8: Recall of DUALTEST (RLB) on over one million samples from the BBC, The Guardian, and The New York Times.

## 5 CONCLUSION AND LIMITATIONS

In this work, we introduce DUALTEST, a practical and interpretable membership inference framework for detecting memorization of copyrighted data in large language models. Motivated by the increasing legal scrutiny of LLMs and the lack of feasible auditing tools, we define four critical criteria for practical memorization tests: black-box access, high precision, efficiency, and interpretability. DUALTEST satisfies all four by leveraging a dual-model setup that contrasts the outputs of a large suspect LLM with those of a smaller reference model. We present two complementary detection strategies—run-length-based and edit-similarity-based—and show that our approach operates with a single query per sample while achieving robust performance under stringent precision constraints. Through controlled experiments on open-source models with known training set as well as large-scale experiments and case studies on models like GPT-4, we demonstrate the reliability and utility of DUALTEST.

**Limitations.** While DUALTEST identifies likely memorization, it does not establish a formal causal link between the training data and model outputs—an important distinction in legal contexts. The reference model's output probabilities must be thresholded carefully: a weak model may assign lower probabilities than even a non-memorizing larger model, leading to underestimation. To mitigate this, we recommend using a low threshold, which makes the test more conservative.

Our method assumes access to a reasonably capable small reference model that has not memorized content to the same extent as the larger (target) model. Ideally, the reference model itself has not memorized the tested samples; if it has, some true cases may be missed (false negatives). This trade-off is acceptable in practice, as prioritizing very low false positive rates is more important than maximizing recall.

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

## APPENDIX

## A  RELATED WORK (CONTINUED)

Early research on membership inference attacks (MIAs) has highlighted privacy concerns (Shokri et al., 2017) and has examined their relation to overfitting (Yeom et al., 2018; Bentley et al., 2020). Sablayrolles et al. (2019) explore connections between black-box and white-box MIAs. Song et al. (2019) bridge the security and privacy domains by analyzing how defenses against adversarial attacks impact MIA effectiveness. Similarly, Kaya & Dumitras (2021) investigate the effects of data augmentation on MIA vulnerability. Defenses to mitigate privacy risks in machine learning have been proposed by Nasr et al. (2018). Additionally, Carlini et al. (2022a) and Zarifzadeh et al. (2023) critically examine standard MIA evaluation methods by emphasizing attack performance at low false-positive rates and proposing stronger attack models. Choquette-Choo et al. (2021) develop MIAs for classification models that rely solely on label information. Lastly, Shejwalkar et al. (2021) are the first to systematically investigate MIAs in text classification.

Extractable memorization, wherein an attacker attempts to extract training data without targeting specific samples, has been examined from a privacy perspective due to the tendency of models to

inadvertently reveal sensitive information, such as personal addresses or messages (Carlini et al., 2019; 2021; Nasr et al., 2023).

Data deduplication methods prove effective in mitigating memorization risks (Kandpal et al., 2022; Lee et al., 2021). To address verbatim memorization during training, defense strategies have been proposed by Hans et al. (2024a).

Membership inference attacks, specifically targeting masked language models, are introduced by Mireshghallah et al. (2022). Several works propose methods for detecting pre-training data based on output probabilities (Shi et al., 2023; Mattern et al., 2023; Oren et al., 2023; Bertran et al., 2023; Zhang et al., 2024b; Zhang & Wu, 2024; Ye et al., 2024; Xie et al., 2024; Wang et al., 2024a; Mozaffari & Marathe, 2024), while others propose output probability-free approaches (He et al., 2025; Kaneko et al., 2024). Song & Shmatikov (2019) develop auditing tools based on training shadow models, and Liu et al. (2024b) utilize models fine-tuned on member data to enhance pre-training data detection. Additionally, Duan et al. (2024) analyze the inherent difficulties of developing MIAs against LLMs, with Puerto et al. (2024) showing enhanced effectiveness of MIAs on longer documents. Recent studies also explore membership inference attacks targeting the context data used by long-context LLMs (Wang et al., 2024b) and Retrieval-Augmented Generation (RAG) systems (Naseh et al., 2025).

Maini et al. (2024) show that many modern MIAs perform no better than random guessing when evaluated carefully. Finally, Zhang et al. (2024a) argue that using statistical tests to prove that a model was trained on a specific data point is unreliable, since it is impossible to sample from the correct null hypothesis—that is, the same model not trained on the target data point.

Various defenses have been proposed to mitigate the generation of copyrighted content, including agent-based approaches (Liu et al., 2024a), model aggregation methods (Abad et al., 2024), and watermarking (Panaitescu-Liess et al., 2025a). Lastly, Panaitescu-Liess et al. (2025b) demonstrate that models can be poisoned to generate copyrighted material, and Liu et al. (2025) show that models can reproduce copyrighted content verbatim from prefixes—even when the specific content was not seen during training.

Another legal concern with the wide usage of LLMs is being able to discern their output from human written text. One example of this closest to our method is Binoculars (Hans et al., 2024b). Binoculars works using two language models to calculate the cross entropy of the input text, identifying how likely the input text is to be generated by the models being used by the detector, flagging anything as likely to have been generated be the detector models as AI generated.

# B  GPT-4 EXPERIMENTS

We evaluate our proposed methods in both normal and adversarial settings.

## B.1  EVALUATION SETUP

**Models.**  Our goal is to assess whether completions from GPT-4 (Achiam et al., 2023) constitute memorization of copyrighted content. We use instruction-tuned models from the LLaMA-3.1 family (Grattafiori et al., 2024)—including 8B, 70B, and 405B variants—as the smaller reference models.

**Datasets.**  We consider completions of articles from Exhibit J in the New York Times v. OpenAI lawsuit (Ravichander et al., 2025) in our main evaluation. As non-member (negative) examples, we use articles from CNN published after the target LLM's knowledge cutoff date, following the experimental setup of Ravichander et al. (2025).

**Baselines.**  We consider similar baselines as in the main section of the paper.

## B.2 RESULTS

We consider both greedy and temperature = 1 sampling for GPT-4 and 50-word prefix (denoted by "Standard" in the tables). For the lawsuit articles, we also consider the prefix from the lawsuit as an alternative.

**Continuous Metrics have Non-Zero Recall, But high FPR.** In Table 9, we tune the continuous-metric methods to achieve 100% precision on normal settings. Although these methods offer high recall (e.g., 7–25%), their FPR remains extremely high (47–98%) in adversarial settings, indicating that they fail to reliably distinguish memorization from generalization. Notably, only one method, Kaneko et al. (2024) w/ zlib, achieves 0% false positive rate. This is because multiplying by the length of the Zlib-compressed text produces a similar effect as our reference model-based method, because repetitive, generalizable text is more compressible. However, as we demonstrate later, this method yields lower recall than ours and lacks interpretability.

**Binary Metrics Tuned for 100% Precision and 0% FPR Fail to Detect Anything.** Table 10 shows the recall for binary-metric methods when tuned to achieve 100% precision in normal settings and 0% FPR in adversarial settings. All the continuous variants have 0% recall, meaning they detect no memorized samples at all. This highlights a trade-off between reliability and utility for these methods.

**Continuous Metrics Tuned for 100% Precision and 0% Recall Also Fail, With One Exception.** Table 11 shows the same analysis for continuous metrics. All methods fail to detect any members under both 100% precision tuning in normal settings and 0% FPR tuning in adversarial settings, except Kaneko et al. (2024) w/ zlib. This method benefits from a weak form of reference: its compressibility heuristic penalizes repetitive but generalizable text, reducing false positives.

**Our Method Outperforms All Baselines.** In Table 12, we compare our method to Kaneko et al. (2024) w/ zlib when tuned for 100% precision in normal settings and 0% FPR in adversarial settings. Our run-length-based method achieves significantly higher recall across some scenarios—e.g., 29% vs. 12% in the "Lawsuit" prefix setting with greedy sampling. Our edit similarity variant achieves slightly higher recall in all settings. Importantly, our methods remain interpretable—unlike methods that use compression-based heuristics such as zlib.

Table 9: **Continuous-metric baselines tuned for 100% precision in normal settings.** Continuous metrics improve recall, but most of them suffer from high FPR (cannot distinguish memorization from generalization).

| Sampling | Prefix | Metric | Karamolegkou et al. (2023) (LCS) | Karamolegkou et al. (2023) (Leven.) | Kaneko et al. (2024) | Kaneko et al. (2024) w/ zlib |
|---|---|---|---|---|---|---|
| Greedy | Standard | Recall | 12% | 7% | 16% | 7% |
| | | FPR | 47% | 97% | 74% | 0% |
| | Lawsuit | Recall | 20% | 17% | 24% | 12% |
| | | FPR | 47% | 97% | 74% | 0% |
| Temperature | Standard | Recall | 13% | 16% | 16% | 13% |
| | | FPR | 47% | 98% | 74% | 0% |
| | Lawsuit | Recall | 25% | 20% | 24% | 23% |
| | | FPR | 47% | 98% | 74% | 0% |

Table 10: **Binary-metric baselines tuned for 100% precision in normal settings and 0% FPR in adversarial settings.** In this case, tuned binary baselines achieve zero recall across all configurations.

| Sampling | Prefix | Kiyomaru et al. (2024)(EM)(+) | Kiyomaru et al. (2024)(BLEU)(+) | Zhao et al. (2024) |
|---|---|---|---|---|
| Greedy | Standard | 0% | 0% | 0% |
| Greedy | Lawsuit | 0% | 0% | 0% |
| Temperature | Standard | 0% | 0% | 0% |
| Temperature | Lawsuit | 0% | 0% | 0% |

Table 11: **Continuous-metric baselines tuned for 100% precision in normal settings and 0% FPR in adversarial settings.** All baselines except for the zlib-based one yield 0% recall.

| Sampling | Prefix | Karamolegkou et al. (2023) (LCS) | Karamolegkou et al. (2023) (Leven.) | Kaneko et al. (2024) | Kaneko et al. (2024) w/ zlib |
|---|---|---|---|---|---|
| Greedy | Standard | 0% | 0% | 0% | 7% |
| Greedy | Lawsuit | 0% | 0% | 0% | 12% |
| Temperature | Standard | 0% | 0% | 0% | 13% |
| Temperature | Lawsuit | 0% | 0% | 0% | 23% |

Table 12: **Comparison of our method to the best baseline (Kaneko et al. (2024) w/ zlib) under 100% precision in normal settings and 0% FPR in adversarial settings.** Our method outperforms the best baseline across all settings. The run-length-based method reaches up to 29% recall and it is more interpretable than the baseline.

| Sampling | Prefix | Kaneko et al. (2024) (w/ zlib) | DUALTEST (RLB) | DUALTEST (ESB) |
|---|---|---|---|---|
| Greedy | Standard | 7% | 26% | 11% |
| Greedy | Lawsuit | 12% | 29% | 22% |
| Temperature | Standard | 13% | 12% | 14% |
| Temperature | Lawsuit | 23% | 21% | 24% |

## C  API COST COMPARISON

We analyze the API cost of DUALTEST in comparison to prior work in Table 13.

Table 13: We analyze the API cost of the original Kaneko et al. (2024) (w/ zlib) method alongside other baselines. Assuming 64 input tokens and 64 output tokens per tested sample, with costs of $0.00003 per input token and $0.00006 per output token (based on current pricing for GPT-4-0613), testing 1,000,000 samples would incur substantial expenses. In comparison, our method is at least 7 times more cost-efficient than these baselines.

| Kaneko et al. (2024) (w/ zlib) $57,600 | Ravichander et al. (2025) $41,400 | He et al. (2025) $255,360 |
|---|---|---|
| DUALTEST (RLB) $5,760 | DUALTEST (ESB) $5,760 | |

## D  EFFECT OF THE REFERENCE MODEL

We test our method with reference models of different sizes (8B, 70B, and 405B) (Figure 3). While the method still works, recall tends to drop as the reference model gets larger, since larger models are more likely to memorize content. Still, having a reasonably strong reference model is important for reliability, as it helps reduce errors in real-world cases.

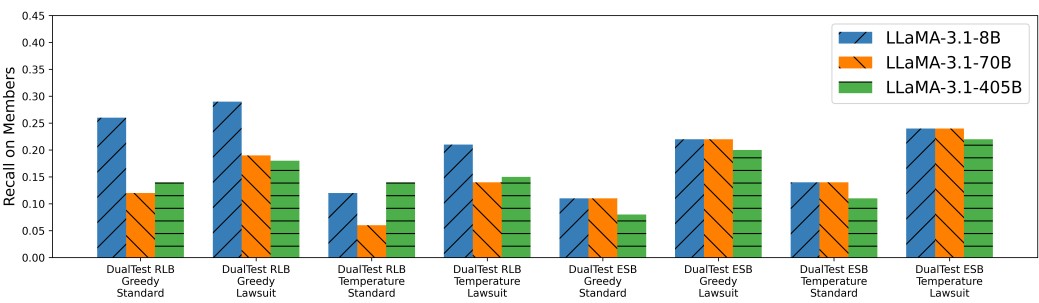

Figure 3: The effect of the reference model on training data detection.

