# OpenReview forum: "Practical Memorization Tests for Detecting Copyrighted Data in Large Language Models"
_ICLR.cc/2026/Conference — ICLR 2026 Conference Withdrawn Submission_

### Official Review · Reviewer_1KzP · 2025-10-20

**Soundness:** 2
**Presentation:** 3
**Contribution:** 2
**Rating:** 2
**Confidence:** 4

**Summary:**

The paper studies the important problem of detecting whether certain pieces of text were used to train an LLM, specifically focusing on the application of detecting the use of copyright protected content. They review the literature of methods that have been proposed and propose a new method, DualTest, that:
- now only relies on black box access to the model,
- is more reliable including against adversarial examples (i.e. samples that are meant to be easy to generalize to, but not included in the training dataset),
- Is efficient (limited number of queries)
- Yields human interpretable results

Authors propose two methods, one based on the length of exactly matched tokens the target LLM produces and one based on the edit similarity between the entire generated sequence and the target one.

They report main results on open source models and also include results for GPT-4. Across setting they report that DualTest achieves high recall at 100% precision, beating all other black box baselines.

**Strengths:**

- The paper studies an important problem, i.e. studying methods to infer whether a certain piece of (copyright-protected) text was used to train a LLM.
- The paper puts the technical methods into the perspective of legal cases, including nicely explained references in the intro, and the emphasis on interpretability of the results.
- The paper focuses on black-box methods, which is important and often overlooked in prior work.

**Weaknesses:**

Overall, I have some doubts regarding how the authors discuss prior work and on the validity of the experimental setup.

**Prior work**:
- While the related work in appendix is more extensive, i find that the summary in the main text can be improved substantially. For instance, from the first paragraph it is not clear what the difference is between MIAs, memorization and extraction, and how this relates to this work. There is a lot of literature on these topics in recent years for LLMs, and clearly articulating these differences is important to understand the paper's contribution.
- Moreover, I find that important related work is missing. For instance, on detecting copyright protected content [1,2,3,4]
- The notion of a reference model is commonly used in MIAs, i.e. originally introduced by Carlini et al. [5] in the same paper that used zlib compression as mentioned by the authors in the introduction. The paper does not properly put their method (section 3.2) into that perspective.

**The validity of the experimental setup**.
-  What subset of MIMIR did the authors use? And can the authors include a bag of words/model-less baseline? This is important, as Meeus et al. [6] find (section VI. A) that the deduplicated subsets of MIMIR also suffer from a distribution shift, for all data subsets for a threshold like 7_0.2 and most notably for github even for other thresholds. This should be properly investigated (i think ideally you use just samples directly from the train/test split from the Pile, or the less aggressive deduplication version, and in any case include a model-less baseline that would need to be surpassed). This will likely also explain the anomalies for github across the paper (e.g. Table 1 and Table 4), as especially for this subset the distribution shift gets very large [6].
- Along the same lines, can authors add a model-less (e.g. bag of words) baseline [6] to all results for GPT-4 and Llama in sections 4.3 and appendix? The data seems to be collected before (members) and after (non-members) a certain cutoff date, a method which is exactly what prior work found to be introducing a distribution shift, also also noted as "evaluation issues' by the authors (lines 150-156). Without a sense whether this data suffers from a distribution shift, we cannot really appreciate these results.
- I like the way authors focus on black box access to the models, but believe that important clarifications and considerations may be missing. For instance, do you use greedy sampling when considering Pythia models as the target? And GPT-4 is an instruction-tuned model that is likely quite different from a pretrained model like Pythia. At least discussing how this would impact the method would be useful. Is there a setup possible where you have access to both the pretrained and the instruction-tuned model as target?
- If authors argue that these methods should be black-box, it is hard to argue that smaller models of the same model family are available (as assumed throughout the main text). It would be helpful to clarify in which cases this is realistic and if it could be added to the limitations.

[1] Rastogi, S., Maini, P., & Pruthi, D. STAMP Your Content: Proving Dataset Membership via Watermarked Rephrasings. In Forty-second International Conference on Machine Learning.

[2] Meeus, M., Shilov, I., Faysse, M., & de Montjoye, Y. A. Copyright Traps for Large Language Models. In Forty-first International Conference on Machine Learning.

[3] Cooper, A. F., Gokaslan, A., Ahmed, A., Cyphert, A. B., De Sa, C., Lemley, M. A., ... & Liang, P. (2025). Extracting memorized pieces of (copyrighted) books from open-weight language models. arXiv preprint arXiv:2505.12546.

[4] Wei, J., Wang, R., & Jia, R. (2024, August). Proving membership in LLM pretraining data via data watermarks. In Findings of the Association for Computational Linguistics ACL 2024 (pp. 13306-13320).

[5] Carlini, N., Tramer, F., Wallace, E., Jagielski, M., Herbert-Voss, A., Lee, K., ... & Raffel, C. (2021). Extracting training data from large language models. In 30th USENIX security symposium (USENIX Security 21) (pp. 2633-2650).

[6] Meeus, M., Shilov, I., Jain, S., Faysse, M., Rei, M., & de Montjoye, Y. A. (2025, April). Sok: Membership inference attacks on llms are rushing nowhere (and how to fix it). In 2025 IEEE Conference on Secure and Trustworthy Machine Learning (SaTML) (pp. 385-401). IEEE.

**Questions:**

- While I believe the use of recall at a 100% precision is a valid metric, I think it is important to also use metrics (even just in appendix) that are more traditionally used to evaluate MIAs, including AUC and TPR at low FPR [1]. Could you report this as well?

- Could authors add a bag of words baseline throughout all results in the paper, including appendix, in line with [2,3]? (see in weaknesses)

I think both of these results would provide a better picture of what the method is actually learning, and to what extent it would be surpassing prior work and model-less baselines.

 [1] Carlini, N., Chien, S., Nasr, M., Song, S., Terzis, A., & Tramer, F. (2022, May). Membership inference attacks from first principles. In 2022 IEEE symposium on security and privacy (SP) (pp. 1897-1914). IEEE.

[2] Meeus, M., Shilov, I., Jain, S., Faysse, M., Rei, M., & de Montjoye, Y. A. (2025, April). Sok: Membership inference attacks on llms are rushing nowhere (and how to fix it). In 2025 IEEE Conference on Secure and Trustworthy Machine Learning (SaTML) (pp. 385-401). IEEE.

[3] Das, D., Zhang, J., & Trantèr, F. (2025, May). Blind baselines beat membership inference attacks for foundation models. In 2025 IEEE Security and Privacy Workshops (SPW) (pp. 118-125). IEEE.

---

### Official Review · Reviewer_pR5U · 2025-10-30

**Soundness:** 2
**Presentation:** 2
**Contribution:** 2
**Rating:** 2
**Confidence:** 3

**Summary:**

This paper introduces a practical memorization test for detecting copyrighted data in LLMs. It proposes four key properties that this test, and other similar methods, must exhibit to be effective in legal cases.
The proposed approach tests whether a language model has memorized copyrighted text by prompting it with a 64-token prefix and then comparing its continuation to the original passage. If the model reproduces the text too closely, and a smaller reference model would be very unlikely to generate the same output, this is taken as strong evidence of memorization.
They evaluate this against other membership inference attacks for LLMs under both a “normal” setting, and an adversarial setting reflecting two likely scenarios in legal cases. The method beats other MIAs by achieving a high precision at a 0% False Positive Rates across datasets and model families.

**Strengths:**

- The paper specifically targets legal applications, designing an MIA that could be used as evidence in copyright disputes which is an interesting and highly relevant subject.
- The paper identifies four key properties relevant to the legal domain: black-box access, high precision, single-query efficiency, and human interpretability. This is interesting as it emphasises the need for the community to agree on practical requirements for real-world applications of such methods. It also provides insight into how previous methods satisfy these properties and where they might fall short.
- The proposed method has two variants that are lightweight and appear to perform well on the task.

**Weaknesses:**

Overall, I find the results somewhat difficult to interpret and not entirely convincing.
- Whilst the threat model is indeed different, I would still think it is important to include relevant baselines from prior work. In particular, simple approaches such as the LOSS attack (or the difference between the target model’s loss and that of a reference model)  [1, 2], as well as Min-K% [3] are missing, making it difficult to interpret the results and overall contribution.
- Previous work has also shown that a bag-of-words classifier can serve as a strong baseline by accounting for distributional shifts between training and test data. This is also necessary to understand the effectiveness of the method [4].
- An ablation study isolating the effects of edit similarity and completion probability would help understand their individual contributions. From Figure 1, it appears that edit similarity alone might already perform quite well.
- While the paper reports precision and recall, this makes it difficult to compare results with previous work, where AUC or TPR@FPR are more standard metrics. Reporting AUC or TPR@FPR would make it easier to assess the trade-offs between true and false positives in a way that is more aligned with established conventions.
- In Section 3.5, the authors note: “We set thresholds on (i) edit similarity and (ii) the reference-model likelihood. This way we can raise precision and reduce false positives.” I however couldn’t find a clear description of how these thresholds were determined and how sensitive the method is to threshold selection across datasets, reference models, etc. Some discussion or analysis here would be quite valuable.
- Importantly, the strong emphasis on achieving 100% precision at 0% FPR seems arbitrary and overly strict. In practice, there is always some tolerance for low false positive rates, especially as the method itself relies on hypotheses which might not be completely true in practice. Results should include precisions at FPRs compared to baselines.
- I find the claim that this method is more interpretable than previous works a bit overstated. Certain approaches such as the LOSS-based tests mentioned above [1,2] are arguably at least as interpretable, if not more so than the proposed method.
- Whilst strong performance under non-privileged access is important, it’d be helpful to provide more justification as to why it is a requirement rather than a feature, especially given the nuances of legal systems
[1] Yeom, Samuel, et al. "Privacy risk in machine learning: Analyzing the connection to overfitting." 2018 IEEE 31st computer security foundations symposium (CSF). IEEE, 2018.
[2] Carlini, Nicholas, et al. "Extracting training data from large language models." 30th USENIX security symposium (USENIX Security 21). 2021.
[3] Shi, Weijia, et al. "Detecting pretraining data from large language models." arXiv preprint arXiv:2310.16789 (2023).
[4] Meeus, Matthieu, et al. "Sok: Membership inference attacks on llms are rushing nowhere (and how to fix it)." 2025 IEEE Conference on Secure and Trustworthy Machine Learning (SaTML). IEEE, 2025.

**Questions:**

- How does the method perform against simpler baselines such as a bag-of-words classifier?
- I’m unclear on requirement (2) in section 3.1 which states “A viable MIA must allow tuning to obtain exceptionally high precision even in adversarial settings”. By “tuning”, do you mean adjusting the threshold to achieve a high True Positive Rate at a specific False Positive Rate? If so, what does the tradeoff curve look like?
- How sensitive is the performance of the method to changes in the threshold on  (i) edit similarity and (ii) the reference-model likelihood, and how are these thresholds determined?

---

### Official Review · Reviewer_R2Yz · 2025-10-31

**Soundness:** 2
**Presentation:** 4
**Contribution:** 3
**Rating:** 4
**Confidence:** 4

**Summary:**

The paper proposes DualTest, a black-box membership inference attack for detecting memorization of copyrighted data in language models. The key idea is to use a small, open-weight reference model to distinguish between memorization and generalization. Crucially, the method supports not making a decision, making it suitable for legal contexts where high precision is essential. The authors evaluate it on both open and proprietary models, showing modest recall gains compared to prior methods while maintaining 100% precision and zero false positives in adversarial settings.

**Strengths:**

* Black-box MIAs against LLMs are difficult, and most prior work relies on open-weight access; tackling the black-box case makes this work valuable and relevant.
* The use of a small open-weight reference model to separate memorization from generalization is a clever and intuitive design for a black-box MIA.
* The paper positions itself well within the current research landscape - they acknowledge and incorporate previous research on approximate extraction, distribution shift in MIAs against LLMs, and prior black-box MIAs against LLMs.
* The method is interesting with a clear intuition why it would work. Indeed, extraction was always the most interesting aspect of attacks against LLMs, but separating memorization from generalization remained the key issue. The proposed method is simple, interpretable, and aligns well with practical constraints (single API call, abstention, interpretability).
* It's great to see authors focusing specifically on extraction of trivial sequences by crafting an adversarial dataset.

Overall I believe this paper has some very interesting ideas, and can be improved to be influential in the field. Some work on the robust evaluation is required though.

**Weaknesses:**

* While authors acknowledge the distribution shift issue, and make a solid attempt (better than vast majority of LLM MIA papers) at avoiding it, more experimental details are needed.
	* First, the validity of the results depend on which exact version if MIMIR is used. Meeus et al. (2024) showed (Table 3) that aggressive deduplication adopted in MIMIR 7_0.2 introduced a new distribution shift.
	* Second, for both versions if MIMIR github subset contains distribution shift  (same Table 3 in Meeus et al. (2024)). This potentially explains better results on github in this paper.
	* Finally, to avoid all doubts on the distribution shift, authors should include blind baselines on the datasets they use.
* Tables 1–5 fix thresholds for 100% precision, but this obscures trade-offs and practical performance. A threshold-independent curve (e.g., log-scale TPR–FPR) would better illustrate model behavior and overall utility, especially when recall is below 1%.
* Table 3 is confusing: FPR is computed on a different dataset than the one used for precision tuning, but this is not clearly explained. The cross-dataset trade-off between TPR and FPR deserves explicit discussion.
* In Table 8, it is unclear how thresholds are set for GPT-4 and LLaMA-2-70B, since true positives and precision are unknowable in those cases.

While I do believe the underlying idea of the method is very promising, current evaluations leave open questions as to how well it actually works.

I will be open to increasing my score if:

a) the results hold on MIMIR subsets without distribution shift (MIMIR 13_0.8, github excluded)

b) results behave as expected when 100% precision threshold is slightly relaxed

**Questions:**

* The method relies on an implicit assumption that a reference model was not trained on a target sample. You discuss this in the limitations section, and assume smaller model would just memorize less. Wouldn't a better option be to use a fully open source model with known training set (like Pythia) and therefore no limit on the model size?
* For run-length-based method it would be interesting to plot run lengths as a second dimension

---

### Official Review · Reviewer_TAMB · 2025-11-01

**Soundness:** 2
**Presentation:** 2
**Contribution:** 1
**Rating:** 2
**Confidence:** 5

**Summary:**

This paper introduces a memorization test for detecting copyrighted material. The authors propose 4 desiderata: black box access, reliability (high precision), efficiency (able to run on a large corpus), and interpretability. The authors describe a method called DualTest, which uses a target LLM to complete text, determines a text based metric (such as the run length matching a copyrighted sample or edit similarity), and normalizes this with the score of a reference model. The authors conduct experiments primarily relying on pythia, gpt4, using the MIMIR benchmark (Duan et al).

**Strengths:**

- Reasonable motivation with desiderata for a MIA test, though does not adequately treat related work.

- Adversarial evaluation is also well motivated as counterexamples can be very valuable in this space.

- Large experimental setup, extensive section of related work though not all experimental methods are discussed further.

- Uses MIMIR to avoid distribution shift issues that are common in earlier work.

**Weaknesses:**

This paper has some interesting experiments but overclaims and does not address key parts of related work, even though those related works are cited throughout and even used as the primary benchmark. There are also some issues with presentation of the existing results which undermines confidence. Therefore it does not yet seem suitable for publication.

W1. This work greatly overclaims novelty. The key idea is to use a smaller reference model, contrasted with methods that use zlib based compression. However, Carlini et al. 2021 explore this idea while many other papers such as Duan et al. 2024 further explore this in the context of the MIMIR benchmark. **This paper focuses primarily on zlib based methods without addressing the extensive existing use of small reference models in other literature.** The two works previously mentioned are cited extensively throughout the paper, and in fact their results use the MIMIR package from that same paper. Thus it is extremely concerning to take such a pick and choose approach to sections of prior work.

W2. This work tries to distinguish itself from other membership inference attack methods (MIA) by stating that their method is more interpretable (214-216):

> 4) Interpretability. For use in legal contexts, the detection criteria must be clear, transparent, and accessible to non-expert audiences (e.g., a judge). Complex metrics lacking straightforward interpretability (such as perplexity or BLEU scores) undermine the practical viability of a test.

The authors claim that perplexity and BLEU score would "undermine the practical viability" of the test, **however they use probability under a reference model which is exactly a form of perplexity.** Further, they propose edit similarity as another part of their method, which is comparable in complexity to BELU score - and surely more complex than a simple BLEU relaxation to something like n-gram precision. This argument does not make sense.

W3. 159-160 the authors state: "In contrast, our work is the first to design MIAs that meet all four practical requirements", where Interpretability is one of the 4 requirements (along blackbox access, reliability, and efficiency). This is wildly overclaimed, as shown by W2 above and analysis of related work. For example, the simple reference model approach described in Carlini 2021: "An alternate strategy is to take a much smaller model trained on the same underlying dataset..." satisfies all of these - reliability can be adjusted through thresholds or careful choice of reference model.

W4. Related, this work uses pythia models as both the test and reference model. However in 471 through 473, the authors state:

> Our method assumes access to a reasonably capable small reference model that has not memorized content to the same extent as the larger (target) model. Ideally, the reference model itself has not memorized the tested samples; if it has, some true cases may be missed (false negatives).

The entire goal of the Pythia suite is to train on identical data for all models - so by their own statement **the authors have picked a poor reference model.** Compare to Duan et al which explicitly experiments with multiple reference models (3.1 of their work).

W5. Tables show results where systems are tuned for 100% precision (the authors state this as a focus of their work). **However the tables show only point-wise results on the precision recall curve** (e.g. for a threshold). This seems biased - standard practice would be to show tradeoff curves for different models such as an ROC graph. Note that recall rates are often reported as 0.0% in tables, or other very low values.

W6. Could consider whitebox methods as an upper bound. Since pythia models are pile based, simple n-gram statistics or an index could be used. Relatedly, the generalization sets proposed are verified by a web search for exact matches (282-283). This is not sufficient verification especially when the exact underlying data can be downloaded and checked directly.

Carlini 2021: https://arxiv.org/pdf/2012.07805

Duan 2024: https://arxiv.org/abs/2402.07841

**Questions:**

- Can you replace tabular results with P/R tradeoff curves rather than single pointwise evals?

---

### Note · Authors · 2025-12-04

**Comment:**

We thank all reviewers for their thoughtful feedback and the time spent evaluating our submission. The comments surfaced several important concerns regarding evaluation design, baselines, and interpretability claims. We appreciate these insights and agree that substantial revisions are needed before the paper is ready for reconsideration.

We would also like to clarify a recurring point raised in multiple reviews: works such as Carlini et al. 2021 and Duan et al. 2024 operate in a **white-box** setting—where the loss on the tested sample or token-level probabilities are available. Our work focuses specifically on the **black-box** scenario, where such information is not accessible. This is why we relied on the zlib-based approach of Kaneko et al. as a baseline rather than methods requiring white-box access.

In future revisions we will address the concerns raised about related work, add the baselines suggested by the reviewers, improve the evaluation metrics, revise our interpretability claims, address distribution-shift concerns, and provide a clearer discussion of assumptions about reference models.

Given the scope of changes required, we have decided to withdraw the submission for now and plan to resubmit once the paper has been substantially improved.

Thank you again for the constructive feedback.

**Withdrawal Confirmation:**

I have read and agree with the venue's withdrawal policy on behalf of myself and my co-authors.